# Survey of Weed Management Practices in Direct-Seeded Rice in North-West Cambodia

**Robert Martin** [1,*]**, Sokunroth Chhun** [2]**, Sophea Yous** [3]**, Ratha Rien** [3]**, Chariya Korn** [3] **and Pao Srean** [3]

[1]  Agricultural Systems Research Co., Ltd., Battambang 02352, Cambodia
[2]  Provincial Department of Agriculture, Forestry and Fisheries, Battambang 02352, Cambodia; chhunsokunroth@gmail.com
[3]  Faculty of Agriculture and Food Processing, National University of Battambang, Battambang 02352, Cambodia; sophea702@yahoo.com (S.Y.); ratharien007@gmail.com (R.R.); kornchariya@gmail.com (C.K.); pao.srean@ubb.edu.kh (P.S.)
*  Correspondence: asrcambo@gmail.com

**Abstract:** Two-hundred rice farming households from eight lowland rice villages in North-West Cambodia were surveyed in 2020 to determine changes in farmers' knowledge, weed management practices and weed seed contamination in seed kept for sowing. The major yield constraints cited by farmers were lack of water, inability to manage water and competition by weeds. Water shortages result in reduced crop establishment, non-optimal herbicide application timing and poor weed control. Reduced tillage, drill planting and use of pre-emergence herbicides can improve weed management. The adoption of drill planting improves crop establishment and enables the use of pre-emergence herbicides. Possible reasons for changes in weed problems include the change to a two-crop rice system in the wet season and spread of seeds by harvesting machines. The main weed seed contaminants of rice seed kept for sowing were *Echinochloa crus-galli*, *E. colona*, *Fimbristylis miliacea*, *Ischaemum rugosum* and *Melochia corchorifolia*. *F. miliacea* was the only species effectively removed by village cleaning methods. Although prevalent in rice fields, *Cyperus. iria* with <0.2 seeds per 500 g and *Leptochloa chinensis* with nil contamination were insignificant contaminants of seed samples. The majority of farmers in the study area are relying on repeated use of a narrow range of post-emergence herbicides, thus leading to increased severity of weed problems in dry direct-seeded rice. Integrated weed management is required to reduce over-reliance on post-emergence herbicides. This will require engagement with the local input supply network to introduce advice on improved weed management and sustainable herbicide use.

**Keywords:** direct-seeded rice; weed management; weed population shifts; selective herbicides; integrated weed management; agricultural mechanisation

## 1. Introduction

Broadcast wet or dry direct-seeding of rice (DSR) is the dominant rice cultivation method in rainfed systems in Battambang and in some parts of Pursat and Banteay Meanchey provinces in North-West Cambodia [1]. The high cost of labour for establishing and managing a seedling nursery plus transplanting was seen as an important reason for adoption of dry direct-seeded rice in the region.

A detailed description of historical direct-seeding practices in Battambang province has been given in [2,3]. This includes "mid-season tillage" for weed management which involves shallow ploughing of direct-seeded rice fields during the tillering stage, usually around August. This buries weeds, especially sedges, and leaves rice plants bent over but not buried. Mid-season tillage also reduces rice plant density where rice is broadcast at excessive rates of 200 kg ha$^{-1}$ to suppress weeds. Mid-season tillage is only practiced in medium and late maturing varieties which have more time to recover from the disturbance and produce more tillers compared to short-duration varieties.

Village Chiefs in Battambang province were interviewed in 2004–2005 [4] and >80% of rice was direct-seeded (broadcast) at that time. Paddy yields were around 2 t ha$^{-1}$ and according to Village Chiefs, direct seeding dates back to before 1975 in Battambang province [4]. By 2017, almost 100% of rice fields in North-West Cambodia were sown by hand-broadcasting [5,6]. Most farmers use their own saved rice seed for sowing to reduce cost. This poses a high risk of weed seed contamination in rice seed kept for sowing. Increased reliance on herbicides in rice has been associated with the transition to wet or dry direct-seeded rice. This has resulted in shifts towards more difficult-to-control weeds and the potential for the evolution of herbicide resistance [7].

In the 1990s, 70% of Battambang rice farmers used in-crop ploughing (mid-season tillage) as the predominant weed control measure and only 15% used herbicides, predominantly 2,4-D [8]. By 2017, 100% of farmers used herbicides for weed control in rice in Battambang province. Results of a survey in 2017 were reported where the majority of farmers (53%) said they changed herbicides every year, 18% changed herbicides every two years, 14% seasonally and 14% did not change herbicides for longer periods [6].

In 2017, the most commonly used selective herbicide in rice was still 2,4-D (76%) and 18% of farmers used 2,4-D as the only herbicide. Other herbicides used were bispyribac-sodium (32%), pyribenzoxim (27%), fenoxaprop + pyrazosulfuron + quinclorac (26%), propanil + clomazone (9%) and bensulfuron + quinclorac (2%). Only 9% of farmers used herbicides outside herbicide mode of action (MoA) groups inhibitors of acetyl CoA carboxylase (Group 1), inhibitors of acetolactate synthase (Group 2) and synthetic auxins (Group 4) [9]. No pre-emergence herbicides were used. In 2020, it was found that post-sowing pre-emergence herbicides such as butachlor and oxadiazon can improve weed control in dry direct-seeded rice in in the region surveyed [5]. These herbicides would provide more timely weed control as well as having different modes of action to the herbicides currently in use.

In 2017, farmers ranked weed species in their rice fields in the following order of importance: *Fimbristylis miliacea* (73%); *Echinochloa crus galli* (56%); *Cyperus iria* (41%); *Leptochloa chinensis* (37%) and *E. colona* (13%) [6]. When asked whether herbicide use increased over time, most farmers said they were increasing their use of herbicides (93%) while 5% said herbicide use was not changing and 2% said herbicide use was decreasing. A recent study [10] provided evidence that Cambodian farmers are locked into a network of neighbors, sellers of pesticides (including herbicides) and company extension staff that reinforce dependence on pesticide use. The findings by [10] reinforce concerns about over-dependence on herbicides, with regard to shifts in species composition and potential evolution of herbicide resistance in weed populations in DSR [7]. About 83% of farmers in the 2017 survey thought that some weed species were evolving herbicide resistance. However, farmers were unable to nominate a weed species that was no longer controlled by a particular herbicide. The main reason for use of herbicides cited was efficacy (74%) but the cost of labor for hand weeding was mentioned by 29% of farmers. The main weed management issues cited by farmers were the high cost of labor and herbicides (68%), lack of knowledge (62%) and misuse or misapplication of herbicides (45%).

The objectives of this study were to:

1. Document rice varieties grown, crop establishment method, source of seed for sowing, paddy yields and losses caused by weeds.
2. Develop an understanding of farmers' weed management practices and decision making processes.
3. Identify emerging weed problems and directions for further research and development for improved weed management in dry direct-seeded rice in North-West Cambodia.

## 2. Materials and Methods

### 2.1. Socio-Economic Characteristics

A survey was conducted in eight communes in Battambang province in North-West Cambodia between May and September 2020 to document farmer knowledge and practices

for weed management in direct-seeded rice systems. A total of 200 households was sampled with 25 from one village in each of eight communes: Bay Damram; Kampong Preang; Ou Mal; Phnum Sampov; Reang Kesei; Preaek Norint; Roka and Ta Kream (Figure 1). Population characteristics of target communes and villages were sourced from a Commune database [11] (Table 1).

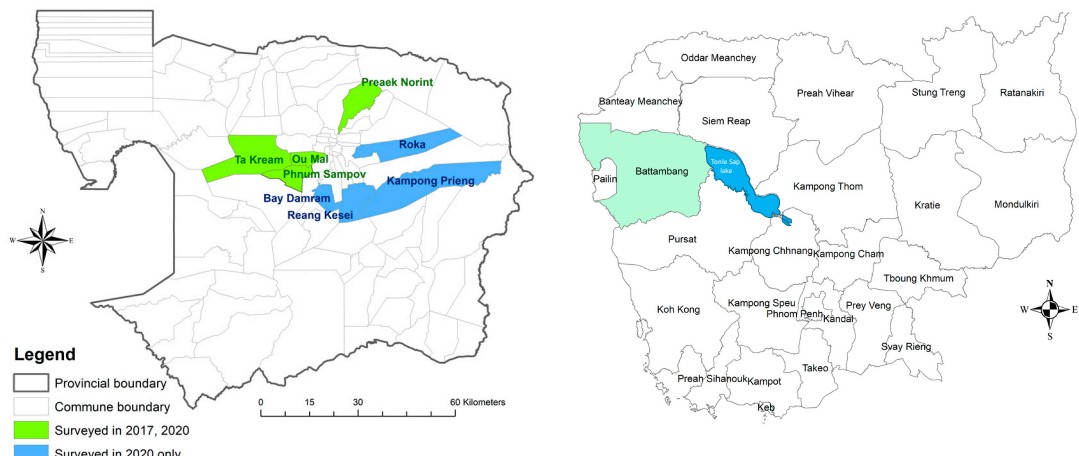

**Figure 1.** Location of communes surveyed. The areas shaded green were surveyed in 2017 and 2020 and the ones shaded blue were surveyed in 2020 only.

**Table 1.** Population characteristics of target communes and villages [11].

| District | Commune | Village | Commune Population | Commune Households | Rice HH [2] Commune | Rice HH (%) |
|---|---|---|---|---|---|---|
| Aek Phnum | Preaek Norint | Rohal Soung [1] | 13,606 | 3010 | 1856 | 62 |
| Banan | Bay Damram | Prey Totueng | 7326 | 1613 | 1246 | 77 |
| Banan | Phnum Sampov | Kampov [1] | 15,873 | 3381 | 2350 | 70 |
| Banan | Ta Kream | Ou Ta Nhea [1] | 20,675 | 3913 | 3111 | 80 |
| Battambang | Ou Mal | Boeng Reang [1] | 10,720 | 2286 | 1544 | 68 |
| Sangkae | Kampong Preang | Os Tuk | 11,534 | 2121 | 1849 | 87 |
| Sangkae | Reang Kesei | Svay Cheat | 8339 | 1851 | 1548 | 84 |
| Sangkae | Roka | Ta Haen Muoy | 10,063 | 1926 | 1308 | 68 |
| | | | 98,136 | 20,101 | 14,812 | 74 |

[1] Also surveyed in 2017. [2] Rice HH = households whose primary occupation is rice farming [11].

Sixty-three percent of interviews were conducted with the household head, 44% of interviewees were female household members and 18% of household heads were female. Twenty-eight percent of households were members of Government-registered Agricultural Cooperatives (ACs) but membership of ACs varied between villages: Svay Cheat (68%); Ou Ta Nhea (56%); Os Tuk (44%); and Prey Totueng (20%). There was a savings group in Ta Haen Muoy village but this was not a registered AC. There was no AC in Boeng Reang, Kampov, or Rohal Soung villages.

The average annual rainfall for Battambang is 1306 mm recorded at the Battambang Department of Water Resources and Meteorology weather station (13.090 N, 103.214 E). Rainfall in 2018 was close to average (1258 mm) but below average in 2019 (1033 mm) with substantially below average rainfall in April, May and June as well as in October and November.

*2.2. Survey Details*

The villages in Ou Mal; Phnum Sampov; Preaek Norint and Ta Kream were also surveyed in 2017 (Figure 1) [6]. Baseline data were collected on methods of land preparation,

varieties, crop yields, access to irrigation, source of sowing seed, planting method and seeding rates.

Farmers were asked about the rice yield reduction caused by weed competition in their fields (after implementation of control measures). They were asked to rank the importance of major sources of weeds occurring in rice fields. They were also shown photos of weeds commonly infesting rice in the region and were asked to list the major weed species present in their rice fields and to rank them in order of importance.

Questions on weed control method and herbicide use included: hand weeding; herbicides used and application timing; water management during herbicide application; satisfaction with the performance of herbicides; weeds not controlled by herbicides. Questions were asked about choice of herbicide and source of information on herbicide choice and application method.

Farmers were asked about changes in weed problems over the last five years: which weeds are becoming more problematic; trends in use of herbicides for weed control; reasons for changed use of herbicides; effective method of weed control; labor scarcity and expense of hand weeding. Farmer observations were sought on weed species previously controlled by herbicide but now not controlled or poorly controlled.

Information was sought on weed management issues in rice fields such as cost, availability of water, labor shortage, lack of knowledge of weed management, climate variability and appropriate use of herbicides. Farmers were asked whether weedy rice occurred in their rice fields and how they ranked its importance. They were also asked if they cleaned rice seed kept for sowing and what method was used.

Rice paddy samples kept for sowing of the 2020 crop were collected from each of the 200 households (25 per village) and inspected for the presence of weed seeds. Samples of approximately 250 g were collected from each household and farmers were asked if seed had been cleaned or not cleaned. Numbers of seeds of each weed species were recorded from 100 g sub-samples. Weed seeds were separated from the rice seeds by hand and inspected with a HOT S06 digital microscope with magnification up to 200× [6]. For identification, weed seed specimens were compared with a collection of HOT seed images of 186 weed species commonly occurring in Cambodia (R. Martin unpublished).

*2.3. Statistical Analysis*

Data were analyzed using IBM$^{©}$ SPSS$^{®}$ Version 22. Discreet variables were analyzed using the Pearson Chi-Square test for association between variables such as farmer scores. Continuous variables such as yield data were analyzed using ANOVA and the means compared using Duncan's Multiple Range post-hoc analysis [12].

The exploitable yield gap (EYG), [13], was calculated using the equation: $EYG = EYf - FY$, where EYf is the mean yield of the top decile farmers (average yield obtained by farmers above the 90th percentile). FY: mean yield of the full sample size.

Redundancy Analysis (RDA) was conducted using the 'vegan' package [14] in R statistical software [15] to analyze the relationship between occurrence of weed species and village location for number of seeds for each species found as contaminants in paddy seed samples and for number of farmers reporting each weed species as problematic in their rice fields. Results are presented as biplot charts.

**3. Results**

*3.1. Agronomy and Rice Production Methods*

The average area of rice owned or managed per household was 5.3 ha and analysis of variance recorded no significant differences between villages. However, farmer estimates of rice paddy yields differed significantly between villages within season (Table 2). The average yield for the first crop was 3.3 t ha$^{-1}$ and 2.4 t ha$^{-1}$ for the second crop (Table 2). In 2018, the average yield for the first crop was 3.5 t ha$^{-1}$ and second crop was 2.7 t ha$^{-1}$. However, in 2019, lack of rain and lack of access to irrigation led to low yields and crop

failure for the second crop, particularly in Boeng Reang, Kampov and Os Tuk whereas in Ou Ta Nhea yields were lower for the first crop in 2019.

**Table 2.** Rice yields in 2018, 2019 and the average for the two years (t ha$^{-1}$).

| Village | 2018 | | 2019 | | Average | |
|---|---|---|---|---|---|---|
| | Season 1 | Season 2 | Season 1 | Season 2 | Season 1 | Season 2 |
| Boeng Reang | 3.44 ab [1] | 2.48 ab | 3.24 bc | 1.69 b | 3.34 bc | 2.08 b |
| Kampov | 3.40 ab | 2.62 ab | 3.32 bc | 0.63 a | 3.36 bc | 1.63 b |
| Os Tuk | 3.55 b | 2.82 b | 2.88 ab | 0.66 a | 3.21 abc | 1.74 b |
| Ou Ta Nhea | 3.48 b | 4.10 c | 2.17 a | 4.74 d | 2.83 ab | 4.42 c |
| Prey Totueng | 3.61 b | 2.93 b | 3.96 cd | 3.06 c | 3.79 c | 3.00 c |
| Rohal Soung | 2.86 ab | 1.96 b | 2.36 a | 1.62 b | 2.61 a | 1.79 b |
| Svay Cheat | 4.63 c | 4.26 c | 4.34 d | 3.84 c | 4.49 d | 4.05 c |
| Ta Haen Muoy | 2.68 a | 0.28 a | 3.22 bc | 0.15 a | 2.95 ab | 0.21 a |
| Average | 3.46 | 2.68 | 3.19 | 2.05 | 3.32 | 2.37 |

[1] Means within columns with the same letter are not significantly different [12].

Ta Haen Muoy village is rainfed only, so a second crop is not expected except for a small number of farmers who do have access to irrigation (Table 2). The overall average exploitable yield gap (EYG) was 1.3 t ha$^{-1}$.

Seventy percent of households had access to scheme irrigation and the main availability was from June to August. The majority of households in Boeng Reang (88%), Kampov (92%), Os Tuk (96%), Ou Ta Nhea (84%), Prey Totueng (96%) and Svay Cheat (96%) had wet-season irrigation. Ou Ta Nhea (36%) also had access to dry season irrigation in January-February. Rohal Soung had limited access to irrigation from the Sangkae River and Ta Haen Muoy had no access to irrigation.

Farmers planted six main varieties of rice. The most popular variety was Sen Kra Oub, planted by 73% of farmers. Other popular varieties were Srangae (24%); Neang Khon (13%); Kinyei Tou (12%); Malise (9%); and Phka Rumduol (5%). Sen Kra Oub was the most popular variety in all villages except Ou Ta Nhea and Ta Haen Muoy.

Srangae was the main variety grown in Ou Ta Nhea village with a small amount in Os Tuk and Prey Totueng villages. Kinyei Tou was grown in Svay Cheat and Prey Totueng villages. Malise was grown by 64% farmers in Rohal Soung village where a short-duration variety is used to enable planting before flooding begins in August. Only 5% of farmers grew Phka Rumduol export Jasmine variety, mainly in Ou Ta Nhea village. Neang Khon, a long-duration variety, was the most popular variety in Ta Haen Muoy village where there is no irrigation and only one crop per year can be grown.

The primary source of seed for sowing is the farmer's own seed (74%) and that bought from other farmers (28%). Negligible seed was sourced from local seed producers (7.5%), Government (1.5%) and input sellers (1%). Seventy-three percent of farmers said they cleaned their own seed with 54% doing their own winnowing and 19% having sowing seed professionally cleaned in the village at a median cost of $38 t$^{-1}$. Cost of village seed cleaning ranged from $25 t$^{-1}$ to $63 t$^{-1}$. Seed cleaning costs increase if the sample is put through the cleaning machine more than one time to obtain a cleaner sample.

The most active villages for seed cleaning were Os Tuk, Svay Cheat and Prey Totueng. The average reduction of weed seed contaminants in sowing seed achieved by farmer seed cleaning was 54%. The most effective seed cleaning was achieved in the three villages which had accessed commercial seed cleaning services: Svay Cheat (89% reduction); Prey Totueng (86% reduction); and Os Tuk (79% reduction). However, the reason for cleaner seed in Rohal Soung village is not yet known.

The majority of farmers (91%) hand-broadcast seed with 6% now planting with a seed drill machine, 2% planting with Eli air seeder [16] and 1% planting with a locally manufactured broadcasting machine. The average seeding rate for hand-broadcasting was 192 kg ha$^{-1}$ for the first crop and 196 kg ha$^{-1}$ for the second crop.

### 3.2. Weed Management Practices and Decision Processes

Most villages were neutral on the importance of weed seed contamination in sowing seed but weed seed contamination was not seen as important in Ou Ta Nhea, Svay Cheat and Ta Haen Muoy villages (Table 3). Most villages recognized the soil seedbank as an important source of weeds but Prey Totueng and Svay Cheat villages regarded the soil seedbank as of lesser importance. The majority of farmers did not regard irrigation water or transfer on ploughs as an important source of weed dispersal. Almost 50% of farmers recognized harvesting machines as an important means of dispersal of weed seeds from field to field and from farm to farm and this was consistent across villages. A minority of farmers recognized transfer of weed seeds by humans or animals as a problem except in Ou Ta Nhea village where this source was seen as important by 40% of farmers.

**Table 3.** Farmers' opinion on the sources of weeds germinating in their rice fields (% of households).

| Source | Soil | Harvest Machine | Sowing Seed | Wind | Irrigation Water | People, Animals | Plough |
|---|---|---|---|---|---|---|---|
| Boeng Reang | 60 | 36 | 48 | 12 | 12 | 16 | 0 |
| Kampov | 84 | 52 | 40 | 20 | 16 | 32 | 8 |
| Os Tuk | 68 | 52 | 24 | 28 | 32 | 4 | 0 |
| Ou Ta Nhea | 56 | 44 | 4 | 16 | 32 | 40 | 16 |
| Prey Totueng | 36 | 48 | 44 | 40 | 8 | 8 | 4 |
| Rohal Soung | 68 | 20 | 44 | 28 | 40 | 20 | 8 |
| Svay Cheat | 40 | 52 | 12 | 20 | 16 | 0 | 0 |
| Ta Haen Muoy | 88 | 48 | 24 | 32 | 12 | 16 | 4 |
| Ave | 64 | 44 | 32 | 24 | 20 | 16 | 4 |
| $\chi^2$ value | 25.9 | 8.8 | 22.5 | 8 | 14.8 | 23.1 | 11.4 |
| Prob. | 0.001 | NS [1] | 0.002 | NS | 0.033 | 0.002 | NS |

[1] NS = not significant.

Ninety-nine percent of farmers considered weeds to be a major problem in their rice fields. Seventy-six percent of farmers estimated net yield losses from weeds after control measures were implemented were in the range 0–20%. However, 24% of farmers considered that yield losses from weeds exceeded 20%. The difference between villages was significant with 48% of farmers in Prey Totueng village estimating yield loss as being greater than 20%. For the remaining villages, 32% of farmers considered that yield losses exceeded 20%.

Overall, 30% of households increased seeding rates during the past five years. The increase was mainly in Svay Cheat and Ta Haen Muoy village where 88% and 84% of households had increased seeding rate respectively. The main reasons for increasing seeding rate were to increase yield (21%), dry seedbed (17%), losses from pests (13%) and to control weeds (12%).

Thirty species were mentioned by farmers as weeds of rice fields and 29 species were found in the paddy kept for planting (Table 4). However, the overall total was 37 species. There were six species found in paddy that were not mentioned by farmers but these were infrequent. The main species mentioned by farmers but not found in significant numbers in paddy were *C. difformis*, *C. iria* and *L. chinensis*. It is assumed that these seeds fall to the ground before harvest or are removed by the combine harvester.

Farmers were asked to rank the major weed species present in their rice fields with the aid of photographs and a total of 30 species were recognized. The most important weed species according to farmers were: *E. crus-galli* (82%); *E. colona* (81%); *C. iria* (60%); *Melochia corchorifolia* L. (60%); *L. chinensis* 59%); *F. miliacea* (50%); *Oryza sativa* f. *spontanea* Baker (34%); *C. difformis* L. (30%); *Cyanotis axillaris* (L.) D. Don ex Sweet (22%); and *Ischaemum rugosum* Salisb. (19%) (Table 5). Species codes used in Table 5 were sourced from EPPO (2020).

**Table 4.** Weed species identified by farmers and recorded in harvested paddy in the survey.

| Family | Species | EPPO Code [17] | Farmer freq (%) | Paddy freq (%) |
|---|---|---|---|---|
| Asteraceae | *Ageratum conyzoides* L. | AGECO | 9 | 1 |
| Commelinaceae | *Commelina benghalensis* L. | COMBE | 1 | 0 |
| Commelinaceae | *Commelina diffusa* N. Burman | COMDI | 2 | 0 |
| Commelinaceae | *Cyanotis axillaris* (L.) D. Don ex Sweet | CYBAX | 22 | 5 |
| Commelinaceae | *Murdannia nudiflora* (L.) Brenan | MUDNU | 0 | 5 |
| Convolvulaceae | *Ipomoea aquatica* Forsk. | IPOAQ | 8 | 0 |
| Convolvulaceae | *Merremia hederacea* (N.L. Burman) H. Hallier | MRRHE | 9 | 4 |
| Cyperaceae | *Actinoscirpus grossus* (L.f.) Goetgh. & D.A. Simpson | SCPGR | 7 | 1 |
| Cyperaceae | *Cyperus difformis* L. | CYPDI | 30 | 2 |
| Cyperaceae | *Cyperus iria* L. | CYPIR | 60 | 1 |
| Cyperaceae | *Cyperus rotundus* L. | CYPRO | 7 | 0 |
| Cyperaceae | *Fimbristylis dichotoma* (L.) Vahl | FIMDI | 4 | 9 |
| Cyperaceae | *Fimbristylis miliacea* (L.) Vahl | FIMMI | 51 | 43 |
| Cyperaceae | *Schoenoplectus juncoides* Roxburgh | SCPJU | 6 | 0 |
| Cyperaceae | *Scleria lithosperma* (L.) Swartz | SCLLI | 2 | 2 |
| Fabaceae | *Aeschynomene americana* L. | AESAM | 1 | 8 |
| Fabaceae | *Aeschynomene indica* L. | AESIN | 4 | 7 |
| Fabaceae | *Aeschynomene aspera* L. | AESAS | 1 | 3 |
| Fabaceae | *Alysicarpus monilifer* (L.) de Candolle | ALZMO | 0 | 5 |
| Fabaceae | *Sesbania bispinosa* (Jacquin) W. Wight | SEBBI | 0 | 2 |
| Linderniaceae | *Lindernia antipoda* (L.) Alston | LIDAP | 2 | 5 |
| Malvaceae | *Melochia corchorifolia* L. | MEOCO | 60 | 22 |
| Malvaceae | *Pentapetes phoenicea* L. | PNPPH | 1 | 3 |
| Onagraceae | *Ludwigia hysoppifolia* (G. Don) Exell | LUDLI | 3 | 2 |
| Phyllanthaceae | *Phyllanthus urinaria* L. | PYLUR | 0 | 1 |
| Poaceae | *Brachiaria mutica* (Forssk.) Stapf | PANPU | 1 | 0 |
| Poaceae | *Cynodon dactylon* (L) Persoon | CYNDA | 1 | 0 |
| Poaceae | *Dactyloctenium aegyptium* (L.) Willdenow | DTTAE | 0 | 1 |
| Poaceae | *Digitaria bicornis* (Lam.) Roemer & Schultes | DIGBC | 0 | 1 |
| Poaceae | *Echinochloa colona* (L.) Link | ECHCO | 81 | 26 |
| Poaceae | *Echinochloa crus-galli* (L.) Beauv. | ECHCG | 82 | 54 |
| Poaceae | *Ischaemum rugosum* Salisb. | ISCRU | 19 | 29 |
| Poaceae | *Leptochloa chinensis* L. | LEFCH | 59 | 0 |
| Poaceae | *Oryza sativa* f. *spontanea* Baker | ORYSA | 33 | 95 |
| Poaceae | *Panicum cambogiense* Balansa | PANCB | 14 | 4 |
| Poaceae | *Panicum repens* L. | PANRE | 0 | 3 |
| Poaceae | *Paspalum scrobiculatum* L. | PASSC | 10 | 15 |

**Table 5.** Farmer ranking of the 10 most important weed species in their rice fields (% of households).

| Village | ECHCG [1] | ECHCO | MEOCO | LEFCH | CYPIR | FIMMI | ORYSA | CYPDI | CYBAX | ISCRU |
|---|---|---|---|---|---|---|---|---|---|---|
| Boeng Reang | 88 | 84 | 40 | 76 | 68 | 48 | 12 | 40 | 0 | 16 |
| Kampov | 76 | 60 | 64 | 68 | 52 | 68 | 52 | 8 | 0 | 12 |
| Os Tuk | 92 | 92 | 52 | 28 | 56 | 36 | 36 | 36 | 44 | 12 |
| Ou Ta Nhea | 84 | 64 | 24 | 72 | 64 | 52 | 24 | 32 | 16 | 40 |
| Prey Totueng | 96 | 96 | 60 | 72 | 64 | 52 | 28 | 48 | 36 | 12 |
| Rohal Soung | 72 | 72 | 72 | 68 | 44 | 48 | 12 | 32 | 0 | 12 |
| Svay Cheat | 100 | 88 | 72 | 60 | 72 | 40 | 40 | 28 | 28 | 24 |
| Ta Haen Muoy | 48 | 92 | 92 | 24 | 56 | 56 | 64 | 16 | 48 | 20 |
| Average | 82 | 81 | 60 | 59 | 54 | 50 | 34 | 30 | 22 | 19 |
| $\chi^2$ value | 33.1 | 21.7 | 32 | 30.63 | 6.1 | 6.7 | 26.6 | 13.7 | 42.6 | 11.1 |
| Prob. | 0.000 | 0.003 | 0.000 | 0.000 | NS [2] | NS | 0.000 | NS | 0.000 | NS |

[1] Species codes [17]. [2] NS = not significant.

Farmer rankings for sedges (Table 5) and *I. rugosum* were not significantly different between villages (Table 5). However, there were significant differences between villages for rankings for most weed species. These differences are likely to be associated with a combination of factors including cropping system, management, soil type and availability of water.

One-hundred percent of farmers use herbicides but 32% still practice hand-weeding. Thirty-one percent of farmers hand-weed once and 23% hand-weed a second time. The timing of hand-weeding varies widely between 18–39 days after sowing in Ou Ta Nhea village to 58–65 days after sowing in Ta Haen Muoy village. The timing of hand-weeding depends on availability of water (Ou Ta Nhea village) and on crop duration such as in Ta Haen Muoy village where mainly long-duration varieties are grown and hand weeding is done later.

Farmers used 15 different herbicide formulations. However, the majority of farmers used a narrow range of post-emergence herbicides: bispyribac-sodium (80%) for control of grasses and sedges and 2,4-D (78%) for control of broadleaved weeds. Fenoxaprop + pyrazosulfuron + quinclorac (23%) was also a popular broad-spectrum post-emergence herbicide. Only 10% of farmers used herbicides other than from Groups 1, 2 or 4 (WSSA, 2020). Thirty-eight percent of farmers have used the same herbicides for more than five years and some have used the same herbicide for up to 10 years. However, 52% of farmers said they change herbicide according to changing weed populations. The median timing of post-emergence herbicide application was from 20 to 30 DAS.

There was significant variation between villages regarding choice of herbicides. The villages engaged in 2017: Boeng Reang, Kampov, Ou Ta Nhea and Rohal Soung use a wider range of herbicides than the additional four villages engaged in 2020 (Os Tuk, Prey Totueng, Svay Cheat and Ta Haen Muoy) (Table 6). Villages engaged in 2017 are less reliant on bispyribac-sodium and 2,4-D compared to the additional villages engaged in 2020.

**Table 6.** Variation in herbicide use between villages: % of households using a particular herbicide.

| Village | Bispyribac-Sodium | 2,4-D | Fenoxaprop + Pyrazosulfuron + Quinclorac | Alternative Herbicides | Using Same Herbicide >5 Years |
|---|---|---|---|---|---|
| Boeng Reang | 76 | 56 | 48 | 60 | 8 |
| Kampov | 56 | 72 | 44 | 56 | 4 |
| Ou Ta Nhea | 40 | 56 | 44 | 72 | 8 |
| Rohal Soung | 84 | 68 | 36 | 60 | 32 |
| Os Tuk | 96 | 100 | 4 | 4 | 64 |
| Prey Totueng | 100 | 92 | 0 | 8 | 60 |
| Svay Cheat | 96 | 80 | 8 | 20 | 64 |
| Ta Haen Muoy | 88 | 96 | 0 | 8 | 80 |
| Average | 80 | 78 | 23 | 36 | 40 |
| $\chi^2$ value | 48.8 | 30.3 | 46.9 | 61.8 | 74.7 |
| Prob. | 0.000 | 0.000 | 0.000 | 0.000 | 0.000 |

Furthermore, herbicide management practices differed between the two village groups. The 2017 group changed herbicides more frequently than the 2020 group and were also more likely to change herbicide depending on the weed species problem rather than scheduled changes.

### 3.3. Emerging Weed Problems and Directions for Further Research

The availability of water and the ability to manage it for weed control is a major constraint to effective weed management in rice in North-West Cambodia. Only 28% of farmers drain water for herbicide application and this is mainly because there is no water in the field to drain. Ou Ta Nhea, in the Kamping Puoy irrigation area, was the only village where the majority of farmers (80%) were able to drain water for post-emergence herbicide

application. Where water is available, the majority of farmers re-flood the field 1–2 days after herbicide application.

The majority of farmers (62%) used motorized knapsack sprayers with an average of 6 nozzles on the boom and 53% contracted others to apply herbicides. The cost of contract herbicide application ranged from $6.00 to $17.50 ha$^{-1}$ with a median cost of $10.00 ha$^{-1}$.

Sixty-seven percent of farmers thought weed problems had increased in the last five years, 22% thought there had been no change and 9% thought weed problems were decreasing. Fifty percent of farmers had not changed herbicide use over the past five years while 43% said they had increased herbicide use.

The main weed species that farmers claim to be not controlled by herbicides are *E. crus-galli* (54%), *E. colona* (37%) and *L. chinensis* (26%). However, many farmers said that lack of water prevented them from applying herbicides at the correct time. Herbicide efficacy was reduced with delayed application and farmers thought these weeds would normally be controlled with timely herbicide application. The same weed species were the main ones considered to be becoming more problematic: *E. crus-galli* (74%); *L. chinensis* (43%); and *E. colona* (39%) (Table 7).

**Table 7.** Weed species that farmers consider to be becoming more problematic (% of households).

| Village | ECHCG | LEFCH | ECHCO | PANCB | CYPIR | FIMMI |
|---|---|---|---|---|---|---|
| Boeng Reang | 88 | 56 | 28 | 0 | 12 | 8 |
| Kampov | 52 | 64 | 4 | 8 | 4 | 0 |
| Os Tuk | 92 | 16 | 72 | 12 | 16 | 4 |
| Ou Ta Nhea | 76 | 76 | 32 | 0 | 24 | 12 |
| Prey Totueng | 88 | 44 | 48 | 0 | 4 | 4 |
| Rohal Soung | 68 | 52 | 12 | 12 | 0 | 12 |
| Svay Cheat | 100 | 20 | 60 | 0 | 20 | 0 |
| Ta Haen Muoy | 32 | 16 | 56 | 56 | 0 | 8 |
| Average | 75 | 43 | 39 | 11 | 10 | 6 |
| $\chi^2$ value | 48.4 | 38.4 | 42.3 | 64.3 | 16.9 | 7.1 |
| Prob. | 0.000 | 0.000 | 0.000 | 0.000 | 0.018 | NS [1] |

[1] NS = not significant.

Sixty-three percent of farmers thought weeds were becoming resistant to herbicides and grasses were the species most commonly suspected of evolving resistance including *O. sativa* f. *spontanea*. However, weedy rice (*O. sativa* f. *spontanea*) is genetically "tolerant" and cannot be controlled by the herbicides being used with the exception of pretilachlor + fenclorim. But only one farmer out of 200 used this herbicide. There are many reasons why a herbicide might fail to control a specific weed species and these should be considered before assuming that herbicide resistance has evolved.

The majority of farmers relied on themselves (85%), the input seller (71%) and the herbicide label (52%) for information on herbicide use (Table 8). The villages least reliant on input sellers for information were Ou Ta Nhea and Ta Haen Muoy. Villages less likely to rely on the herbicide label were Ou Ta Nhea and Svay Cheat. There was some government influence in Ou Ta Nhea and Os Tuk villages but there was no reliance on companies or non-governmental organisations (NGO). This implies that the local supply network is an important pathway to introduce advice on improved weed management and sustainable herbicide use.

**Table 8.** Sources of information on herbicide use (% of households).

| Village | Myself | Input Seller | Label | Company | Govt | NGO |
|---|---|---|---|---|---|---|
| Boeng Reang | 80 | 76 | 56 | 12 | 4 | 0 |
| Kampov | 92 | 76 | 64 | 4 | 0 | 4 |
| Os Tuk | 96 | 64 | 52 | 4 | 16 | 8 |
| Ou Ta Nhea | 80 | 52 | 36 | 8 | 32 | 4 |
| Prey Totueng | 68 | 92 | 68 | 4 | 0 | 0 |
| Rohal Soung | 88 | 92 | 64 | 0 | 0 | 0 |
| Svay Cheat | 84 | 72 | 24 | 8 | 8 | 8 |
| Ta Haen Muoy | 92 | 44 | 52 | 8 | 4 | 8 |
| Average | 84 | 72 | 52 | 8 | 8 | 4 |
| $\chi^2$ value | 11.1 | 25.2 | 6.0 | 4.3 | 29.3 | 6.3 |
| Prob. | NS | 0.001 | 0.025 | NS [1] | 0.000 | NS |

[1] NS = not significant.

The main weed management issues in rice fields were lack of water (98%), lack of knowledge generally (91%) herbicide knowledge (84%) and climate variability (83%) (Table 9). Ou Ta Nhea, in the Kamping Puoy irrigation area, had a slightly lower score for lack of water but lack of water was still nominated as a problem by 88% of farmers. Ou Ta Nhea (76%) also had a lower score for lack of knowledge.

**Table 9.** The main factors affecting weed management in rice fields according to farmers (% of households).

| Village | Lack of Water | Lack of Knowledge | Lack of Herbicide Knowledge | Climate Variability | Labour Shortage | Cost of Labour |
|---|---|---|---|---|---|---|
| Boeng Reang | 100 | 88 | 80 | 68 | 36 | 28 |
| Kampov | 100 | 100 | 84 | 96 | 60 | 40 |
| Os Tuk | 100 | 100 | 96 | 88 | 16 | 28 |
| Ou Ta Nhea | 88 | 76 | 72 | 76 | 56 | 40 |
| Prey Totueng | 100 | 92 | 88 | 88 | 12 | 28 |
| Rohal Soung | 96 | 92 | 84 | 72 | 32 | 20 |
| Svay Cheat | 100 | 84 | 80 | 88 | 24 | 24 |
| Ta Haen Muoy | 100 | 96 | 92 | 92 | 24 | 12 |
| Average | 98 | 91 | 85 | 84 | 33 | 28 |
| $\chi^2$ value | 16.3 | 14.4 | 7.6 | 13.0 | 24.6 | 7.8 |
| Prob. | 0.022 | 0.044 | NS [1] | NS | 0.001 | NS [1] |

[1] NS = not significant.

There were 29 weed species with seeds contaminating rice seed for sowing. Seven species accounted for 92% of weed seed contamination of seed kept for sowing. The species composition of weed seeds contaminating rice paddy varied significantly between villages except for *P. cambogiense* (Table 10). Redundancy analysis was used to identify species groupings that could be identified with differing agro-hydrological characteristics between villages (Figure 2). Ta Haen Muoy, with no irrigation, differed from other villages by having more seeds of *I. rugosum*, *C. axillaris*, *A. americana* and less seeds of *E. colona*, *E. crus-galli* and *P. scrobiculatum*. For Ou Ta Nhea, the only village with access to dry season irrigation, paddy samples were characterised by having above average frequencies of *E. colona*, *P. scrobiculatum* and *M. corchorifolia* and a low frequency of *F. miliacea*. Rohal Soung, the only village in the active floodplain, has no scheme irrigation. Paddy samples were characterized by above average frequencies of *F. miliacea* and below average for *M. corchorifolia*. There were no obvious differences for weed seed contamination of paddy between the five villages with wet-season irrigation (Boeng Reang, Kampov, Os Tuk, Prey Totueng, Svay Cheat).

**Table 10.** Weed seed contamination of rice seed kept for sowing (% of households).

| Village | Weed Species | | | | | | | | | |
|---|---|---|---|---|---|---|---|---|---|---|
| | **AESAM** | **CYBAX** | **ECHCO** | **ECHCG** | **FIMMI** | **ISCRU** | **MEOCO** | **ORYSA** | **PASSC** | **PANCB** |
| Boeng Reang | 4 | 0 | 39 | 74 | 35 | 26 | 13 | 100 | 13 | 9 |
| Kampov | 0 | 6 | 11 | 33 | 11 | 17 | 11 | 100 | 28 | 6 |
| Os Tuk | 0 | 4 | 54 | 58 | 17 | 17 | 0 | 100 | 21 | 0 |
| Ou Ta Nhea | 12 | 6 | 36 | 61 | 6 | 30 | 58 | 100 | 27 | 0 |
| Prey Totueng | 0 | 0 | 35 | 65 | 35 | 20 | 10 | 100 | 15 | 5 |
| Rohal Soung | 0 | 0 | 4 | 63 | 54 | 33 | 4 | 92 | 0 | 4 |
| Svay Cheat | 6 | 0 | 24 | 67 | 21 | 9 | 12 | 82 | 12 | 3 |
| Ta Haen Muoy | 32 | 24 | 0 | 0 | 12 | 76 | 52 | 88 | 0 | 4 |
| $\chi^2$ value | 30.1 | 24.4 | 31.5 | 40.6 | 25.7 | 37.8 | 54.5 | 19.5 | 16.2 | 4.3 |
| Prob. | 0.001 | 0.001 | 0.000 | 0.000 | 0.001 | 0.000 | 0.000 | 0.007 | 0.023 | NS [1] |

[1] NS = not significant.

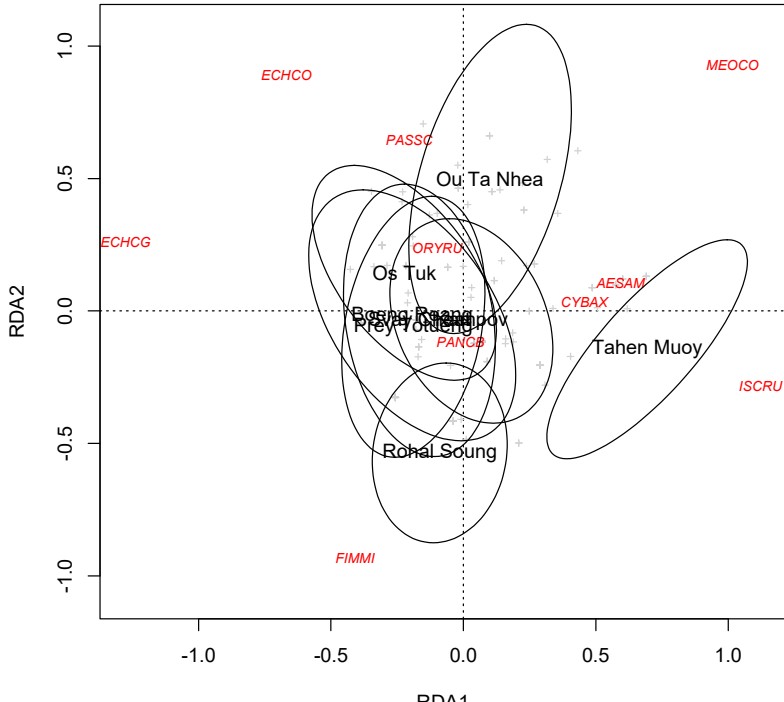

**Figure 2.** Redundancy analysis (RDA) of weed species diversity of contaminants in paddy seed samples.

Frequencies of weed seed contaminants in paddy (Table 10, Figure 2) differed from farmer field observations (Table 5, Figure 3). Weed species with small seeds might be prevalent in rice fields but are rarely found in paddy samples (e.g., *C. difformis*, *C. iria*, *L. chinensis*).

However, weed species with small seeds but retained in inflorescences or fruiting bodies (e.g., *F. miliacea*) tend to be retained in the paddy sample. The redundancy analysis (Figure 3) showed Ta Haen Muoy village to be clearly drawn out by *O. sativa* f. *spontanea* and *M. corchorifolia*. Neighbouring villages, Os Tuk, Svay Cheat and Prey Totueng were clustered together as were neighbouring villages Boeng Reang and Ou Ta Nhea. Rohal Soung and Kampov were drawn out by *F. miliacea*.

The average level of contamination for the total of weed seeds was 795 seeds kg$^{-1}$ with weedy rice accounting for 477 seeds kg$^{-1}$ or 60%. Contamination by *M. corchorifolia* was significantly greater in seed samples from Ou Ta Nhea and Ta Haen Muoy. Weed seed contamination in rice seed for sowing was predominated by weedy rice (*O. sativa* f. *spontanea*: awnless, 445 seeds kg$^{-1}$ and awned 32 seeds kg$^{-1}$ with a total of 477 seeds kg$^{-1}$. *E. crus-galli* was second to awnless weedy rice with 86 seeds kg$^{-1}$. If seeds of rice were to

be sown at 200 kg ha$^{-1}$, at a seed weight of 0.27 g, 741 rice seeds are sown per m$^2$ and with 57% establishment, the established rice plant population would be 424 plants per m$^2$. In comparison, weedy rice contamination at 477 seeds kg$^{-1}$ would result in 9.5 weedy rice seeds being sown per m$^2$ and for *E. crus-galli*, 1.7 seeds would be sown per m$^2$.

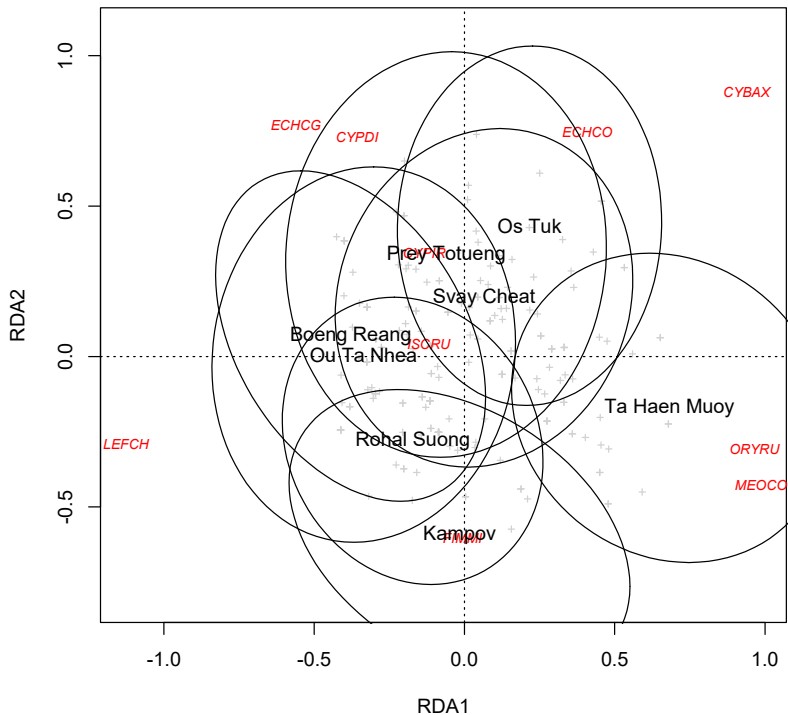

**Figure 3.** Redundancy analysis (RDA) of weed species as reported by farmers in their fields.

## 4. Discussion

### 4.1. Agronomy and Rice Production Methods

The exploitable yield gap for 2018–2019 was 1.4 t ha$^{-1}$ in Boeng Reang, Kampov, Ou Ta Nhea and Rohal Soung compared to 1.3 t ha$^{-1}$ recorded for the same villages in 2016–17 [6].

Average wet season paddy yields for the villages surveyed in 2016–2017 (3.3 t ha$^{-1}$) were similar in 2018–2019 (3.2 t ha$^{-1}$). However, yields appear to have increased slightly since 2008 when the average was 2.86 t ha$^{-1}$ [2]. According to farmers, lack of water and weeds were likely to be the main contributors to the yield gap. If these can be corrected, improved crop nutrition would become a priority to further reduce the yield gap [6].

Since 2008, there has been a shift from the predominance of medium and long duration varieties to short duration varieties. In 2008, the predominant varieties grown were medium and long duration photoperiod-sensitive varieties and short duration varieties were primarily grown under irrigation in the dry season in schemes such as the Kamping Puoy irrigation area [2]. With the completion of the 1st phase of the restoration of the Kanghot irrigation scheme in 2013 [18], there has been an increase in availability of wet season irrigation which allows two short-duration varieties to be grown in the wet season in most villages. This resulted in an increase in short duration varieties from 17% in 2009 to 58% in 2017 [6]. By 2019, 80% of varieties grown were short duration, particularly in villages with access to irrigation. Long duration varieties are still grown in villages such as Ta Haen Muoy where scheme irrigation is not yet available.

The transition to short duration varieties, in combination with machine planting and improved agronomic practices, provide farmers with new options for climate change adaptation in Cambodian rice systems [19], using the APSIM-Oryza crop simulation model [20]. The same model predicted that the optimum sowing month for rainfed rice

in Battambang province is July [21]. However, the farmer-practice median sowing month is now May for villages with wet season irrigation (Boeng Reang, Kampov, Os Tuk, Prey Totueng and Svay Cheat). With wet season irrigation, these villages can also sow a second crop in September (Prey Totueng, Svay Cheat) or October (Boeng Reang, Kampov, Os Tuk). In Ta Haen Muoy village which has no wet season irrigation, rice is planted in July consistent with the modelled optimum. In Ou Ta Nhea village, with access to dry season irrigation from the Kamping Puoy scheme, the median sowing month for the wet season crop is also July which is in agreement with the modelled optimum. January is the median month for sowing the irrigated dry season crop. This is consistent with the recommendation [2] to plant the dry season crop in early January rather than February to avoid anthesis coinciding with the hottest month, April, when temperatures exceed 35 °C resulting in spikelet sterility [22].

Improved water security provided by wet season irrigation has encouraged farmers to plant two short-duration rice crops with the first planted in May. Irrigation water is delivered as early as June but might not arrive until July or August. Water shortages between sowing and when irrigation water arrives result in reduced crop establishment, delayed herbicide application and poor weed control. Reduced tillage, drill planting and use of pre-emergence herbicides can improve water security and weed control in crops sown in May [7].

### 4.2. Trends in Weed Control Practices

One of the objectives of this study was to identify emerging weed problems and management issues that have arisen since the survey in 2017 [6]. Historically, the main method of weed control in direct-seeded rice in the study area was mid-season tillage where medium and long-duration crops are ploughed or harrowed 30–80 days after crop emergence, depending on water accumulation in the field [2]. Mid-season tillage was practiced by 70% of farmers in the 1990s whilst only 15% of farmers used herbicides, mainly 2,4-D at that time [8]. Farmers were still using mid-season tillage in 2008–2009 and this varied between years (18–62%) depending on need and the only herbicide used was still 2,4-D which gave rise to concerns about control of grass weeds such as *Echinochloa* spp. [2].

By 2016–2017, 76% of farmers were using 2,4-D with 18% still using 2,4-D as the sole herbicide [6]. By this time, farmers had begun to diversify herbicide active ingredients with 32% of farmers using bispyribac-sodium, pyribenzoxim (27%), fenoxaprop + pyra-zosulfuron + quinclorac (26%) and propanil + clomazone (9%) [6]. In 2018–19, 100% of farmers still used herbicides but the herbicides used continued to change with use of 2,4-D decreasing by 13% (Figure 4). Use of bispyribac-sodium increased dramatically by 32% and use of fenoxaprop + pyrazosulfuron + quinclorac increased by 17%. This is consistent with the increasing problems with grass weeds, especially *Echinochloa* spp. Use of propanil + clomazone remained the same at 9%.

Although pre-emergence herbicides are considered to be the best herbicide option in dry-seeded rice systems [23], they were not used in the 2017 survey [6] nor in the current survey. Pre-emergence herbicide options for dry direct seeded rice in North-West Cambodia include butachlor, oxadiazon and pendimethalin [21]. However, pendimethalin can be phytotoxic to rice in poorly prepared seedbeds and at higher rates of application [24]. It has been confirmed that butachlor and oxadiazon can improve weed control in dry direct-seeded rice in the surveyed region [5]. However, further research is required to determine the crop safety of pendimethalin, On-farm demonstrations of pre-emergence herbicides in dry direct-seeded rice are warranted in the study area.

Compared with the 2017 survey [6], weed species that have increased in importance according to farmers include *E. colona* and *M. corchorifolia*. In the 2020 survey, farmers mentioned a new species (*Panicum cambogiense* Balansa) not mentioned in the 2017 survey. There are very few references to *P. cambogiense* as a weed of rice in the literature although it is recorded as a weed of deep water rice in central Thailand [25], occasionally flooded meadows in southern Thailand [26], and in upland rice systems in Laos [27]. *P. cambogiense*,

also known as *P. luzonense*, is recorded as occurring naturally in seasonally flooded and burnt grasslands in the middle-outer floodplain of the Tonle Sap lake [28]. The reason for the spread of *P. cambogiense* into rice fields in the older alluvial plains further from the lake is not yet known. However, it is possible that the spread of *P. cambogiense* is associated with the recent changes in rice systems with access to wet season irrigation and rapid transition to short-duration rice varieties in this area. *P. cambogiense* sets seed in September-October and this coincides with harvest of the first wet season rice crop and the potential for spread by harvesting machines.

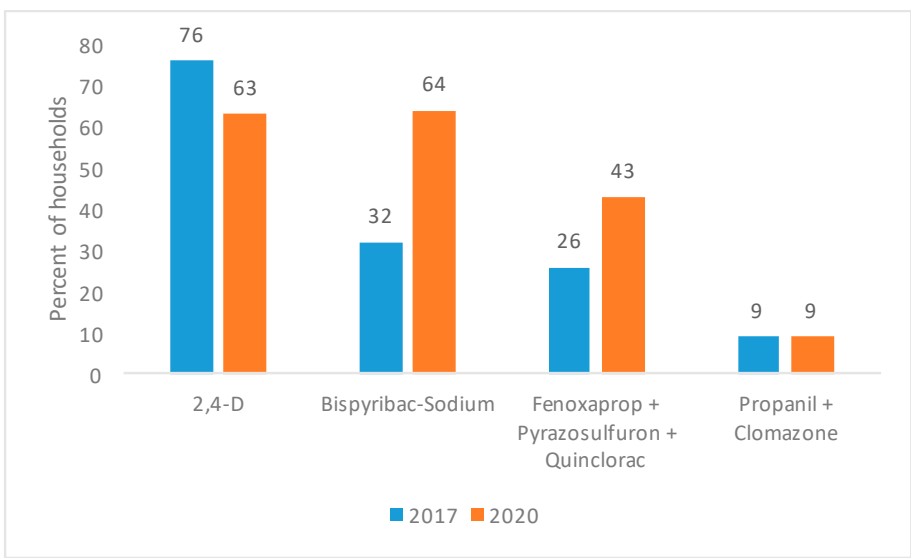

**Figure 4.** Changes in herbicide use in Boeng Reang, Kampov, Ou Ta Nhea and Rohal Soung villages between 2016–2017 and 2019–2020.

Wild rice (*Oryza rufipogon*) is a perennial grass that commonly occurs in in drains and around ponds close to cultivated rice fields and was the predominant species in the seasonally flooded natural grasslands surrounding the Tonle Sap lake before the beginning of modern rice cultivation in the area [28]. Weedy rice is the product of interspecific hybridization between wild rice and cultivated rice [29]. Almost all farmers in the survey were familiar with weedy rice (99%) but only 35% of farmers said that weedy rice was a problem in their fields, However, in the seed contamination study we found that 94% of paddy samples kept for sowing were contaminated with weedy rice with 91% of samples containing seeds of awnless and 57% containing seeds of awned weedy rice. As found in 2017 [6], this suggests that farmers might have difficulty identifying weedy rice, especially awnless biotypes. There is also anecdotal evidence that professionals might be misidentifying awnless weedy rice as varietal off-types.

The level of weed seed contamination in paddy kept for sowing in 2020 was similar to 2017. The only significant difference was for contamination by *E. colona* which was less in 2020. The main weed seed contaminant is awnless weedy rice (*O. sativa* f. *spontanea*). Two villages (Prey Totueng and Svay Cheat) are now employing professional seed cleaning services and these claim to be able to remove at least some weedy rice seeds depending on size and shape differences between weedy rice and the variety.

### 4.3. Integrated Weed Management Options for Dry Direct-Seeded Rice

Farmers recognized the importance of the soil seedbank (64%), harvesting machines (44%) and contaminated seed for sowing (32%) as sources of weed infestations in their rice fields (Table 7). These findings point to a range integrated weed management options that can be considered to improve weed management without significant additional investments. This implies that the local supply network is an important pathway to introduce advice on

improved weed management and sustainable herbicide use. This has been confirmed in other studies [10] but the prevailing response continues to focus on farmers alone through government-based interventions [10]. Our study (Table 8) confirms the need for engagement of all stakeholders who influence farmers' weed management decisions. Changes in weed management, particularly herbicide use, cannot be made without effective engagement with input sellers as well as distributors and importers of herbicides.

## 5. Conclusions

The average rice paddy yield is 3.2 t ha$^{-1}$ and the yield for the top decile farmers is 4.6 t ha$^{-1}$ and the exploitable yield gap (EYG) is therefore 1.4 t ha$^{-1}$. The main constraints to reducing EYG are lack of water, lack of nutrients and yield losses from weeds. Immediately available weed management options to close the yield gap include: stale seedbed; drill seeding; use of clean seed; pre-emergence herbicides; mechanical and hand weeding. The majority of farmers in the study area are relying on repeated use of a narrow range of post-emergence herbicides which have not reduced the weed problem. There is a need to diversify weed management options especially for key grass weeds such as *Echinochloa crus-galli* and *Leptochloa chinensis* and possibly the sedge, *Cyperus iria*. The increasing weed problem in dry direct-seeded rice places priority on validation and implementation of a sustainable integrated weed management strategy. This will require engagement with the local supply network to introduce advice on improved weed management and sustainable herbicide use.

**Author Contributions:** Conceptualization, S.C. and R.M.; methodology, S.C. and R.M.; statistical analysis, P.S. and R.M.; investigation and data processing, S.C., C.K., R.R., S.Y.; writing—original draft preparation, R.M.; writing—review and editing, R.M.; visualization, R.M.; supervision. All authors have read and agreed to the published version of the manuscript.

**Funding:** Funding for this research was provided by the Australian Centre for International Agricultural Research (ACIAR) through the project CSE/2015/044 "Sustainable intensification and diversification in the lowland rice system in Northwest Cambodia".

**Informed Consent Statement:** Informed consent was obtained from all subjects involved in the study.

**Data Availability Statement:** Data are available on request.

**Conflicts of Interest:** The authors declare no conflict of interest.

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
