# Peer review of "Survey of Weed Management Practices in Direct-Seeded Rice in North-West Cambodia"

_agronomy, doi:10.3390/agronomy11030498_

Round 1

Reviewer 1 Report

The manuscript addresses very important issues: methods of rice cultivation in a rainfed system in Southeast Asia, weed management in the rice crop, and farmers' awareness of weed problems and control strategies.

My general suggestions are as follows:

  • Some information placed in Results (lines 142-151) should be moved to Materials and Methods. The latter section is suggested to be divided into subsections on site and population characteristics, survey details, and statistical analysis.
  • I suggest the Authors combine the Results section with the Discussion section and then distinguish the appropriate subsections corresponding to the objectives presented in the Introduction. What is more, baseline data analysis (Lines 152-209) should be presented separately and shortened.
  • It also seems to me a good idea to post the full version of the farmer survey as supplementary material.

Specific comments:

  • Line 130: “…Pearson Chi-Square test for association between variables…” - in some tables the authors include Chi-Square values, but it is difficult to find evidence of test use in the text
  • Line 131: “….yield data were analysed using ANOVA and the means compared using Duncan’s Multiple Range post-hoc analysis” - there is no information about the results of these tests (significant or non-significant differences) in the relevant tables
  • Line 137: “….weeds species diversity…” – please add information on how species diversity was expressed (species number, Shannon entropy ?)
  • How were the averages presented in the tables calculated? Are they arithmetic or weighted averages?
  • Line 168: “The village * year and village * crop interactions were significant” – this significance is not indicated in the Table
  • Line 178: “…Sen Kra Oub (73%)...” – please explain the numbers in brackets when using for the first time in this sentence
  • Pie charts (Figure 4-6) are not appropriate for presenting the results of a multiple-choice survey. The total does not represent 100% here.
  • Lines 211-212 - This interpretation of the data in Table 5 is unclear to me
  • Table 6 – please explain ‘Farmer freq’ and ‘Paddy freq’ under the Table
  • Line 240: (82%) – please explain the number in the first brackets in this sentence
  • Make sure that the conclusions answer the objectives set in the Introduction
  • Tables and figures should be included in the text after they have been cited, not before.

I hope my suggestions will help Authors improve the article in terms of content and form.

Reviewer 2 Report

This paper entitled "Survey of weed management practices in direct-seeded rice in north-west Cambodia" reports weed management status in DSR in NW Cambodia. So, this paper is a kind of milestone paper useful for weed scientists to understand the status of weed management for rice in Cambodia. In this sense, this paper is vauable to be published in Agronomy. However, as this paper contains too much information, which is not well organized in Result section, a significant improvement should be made prior to positive consideration of potential publication in Agronomy.

Please see my comments below

1. Abstract:

- The objective of this survey is unclearly described. I don't think the objective is to determine the changes.

- It is better to focus those findings associated with weed management and those findings need to be more logically rearranged to lead to a clear conclusion.

2. Introduction:

- The reason why this survery and study is required is not well described. The paragraph between lines 74-81 and the objectives from line 82 is not well logically linked. Add some more statements why this survey is needed.

3. Materials and methods

- If possible, it is better to divide current Materials and methods into several subsections such as survey site, survey method, statistical analysis etc.

- In Figure 1, Left map showing communes surveyed is unclear. Too small font size etc.

- In the case of assessing weed seed contamination, how many replications did you test? More detailed information including villages where rice seeds were sampled, rice seed sampling, etc. need to be added.

4. Results

- Too much information and some of them redundant as well --> It is better to rearrange by dividing Result section into a few sub-sections with appropriate sub-section titles representing each context (key finding) like Discussion.

- I think authors were going to divide Results into a few sub-sections considering the first part entitled "3.1. Seeding Method and Seeding Rate". However, there are no following sub-sections such as 3.2...3.3...3.4...etc., thus making Results difficult to follow and understand and resulting in poor readability.

- It is better to minimize Figures and Tables by removing redundant ones, for example, Figure 2 and Table 3 contain the same information. Many of less important ones can be removed or moved to supplementary data, for example, Figures 3 and 5, which are not much relevant to weed management. Figures 10 & 11 contain similar information, so one of them can be removed or moved to supplementary data. This paper has 14 Tables and 12 Figures, which can be reduced to a half of current Figures and Tables.

5. Discussion:

- Current Discussion has 8 sub-sections, too much divided. It is better to have 3~4 subsection with emphesis on those findings associated with weed management. You don't need to list and deal with everything you found from the survey.

6. Conclusion:

- Authors emphesized herbicide resistance too much although this paper reports only 1 Table concerning herbicide resistant weed. It is better to make a final conclusion in direct relation to key findings associated with weed management in rice.

7. Minor and general comments

- Keep consistency in using terms such as rice or paddy

- Figures should be significantly improved. Some of texts in Figures were broken (collapsed), so need to be corrected

- Herbicide MOA: It is better to directly mention mode of action instead of MOA number as Agronomy is not pesticide or weed science journal.

- Keep consistency in using abbreviation such as DSR etc. Full name and abbreviation are mix-used.
